# Aryl Hydrocarbon Receptor-Dependent and -Independent Pathways Mediate Curcumin Anti-Aging Effects

**DOI:** 10.3390/antiox11040613

**Published:** 2022-03-23

**Authors:** Vanessa Brinkmann, Margherita Romeo, Lucie Larigot, Anne Hemmers, Lisa Tschage, Jennifer Kleinjohann, Alfonso Schiavi, Swantje Steinwachs, Charlotte Esser, Ralph Menzel, Sara Giani Tagliabue, Laura Bonati, Fiona Cox, Niloofar Ale-Agha, Philipp Jakobs, Joachim Altschmied, Judith Haendeler, Xavier Coumoul, Natascia Ventura

**Affiliations:** 1Institute of Clinical Chemistry and Laboratory Diagnostic, Medical Faculty, Heinrich Heine University Düsseldorf, Moorenstr 5, 40225 Düsseldorf, Germany; vanessa.brinkmann@uni-duesseldorf.de (V.B.); margherita.romeo@iuf-duesseldorf.de (M.R.); alfonso.schiavi@uni-duesseldorf.de (A.S.); ficox100@uni-duesseldorf.de (F.C.); Niloofar.ALE-AGHA@uni-duesseldorf.de (N.A.-A.); philipp.jakobs@hhu.de (P.J.); joalt001@hhu.de (J.A.); juhae001@hhu.de (J.H.); 2IUF—Leibniz Research Institute for Environmental Medicine, Auf’m Hennekamp 50, 40225 Düsseldorf, Germany; anne.hemmers@hhu.de (A.H.); tschage@em.uni-frankfurt.de (L.T.); jenny.kleinjohann@web.de (J.K.); swantje.steinwachs@iuf-duesseldorf.de (S.S.); charlotte.esser@uni-duesseldorf.de (C.E.); 3Faculté des Sciences Fondamentales et Biomédicales, Université de Paris, 45 Rue des Saints-Pères, F-75006 Paris, France; luciole271@gmail.com (L.L.); xavier.coumoul@parisdescartes.fr (X.C.); 4Institute of Biology, Humboldt-University Berlin, Philippstr. 13, 10115 Berlin, Germany; ralph.menzel@biologie.hu-berlin.de; 5Department of Earth and Environmental Sciences, University of Milano-Bicocca, Piazza della Scienza 1, 20126 Milano, Italy; s.gianitagliabue@gmail.com (S.G.T.); laura.bonati@unimib.it (L.B.); 6Institute of Clinical Pharmacology and Pharmacology, Medical Faculty, University Hospital and Heinrich Heine University Düsseldorf, Moorenstr 5, 40225 Düsseldorf, Germany

**Keywords:** aryl hydrocarbon receptor, curcumin, oxidative stress, *Caenorhabditis elegans*, mice, endothelial cells, in vivo, in vitro, in silico

## Abstract

The aryl hydrocarbon receptor (AhR) is a ligand-activated transcription factor whose activity can be modulated by polyphenols, such as curcumin. AhR and curcumin have evolutionarily conserved effects on aging. Here, we investigated whether and how the AhR mediates the anti-aging effects of curcumin across species. Using a combination of in vivo, in vitro, and in silico analyses, we demonstrated that curcumin has AhR-dependent or -independent effects in a context-specific manner. We found that in *Caenorhabditis elegans*, AhR mediates curcumin-induced lifespan extension, most likely through a ligand-independent inhibitory mechanism related to its antioxidant activity. Curcumin also showed AhR-independent anti-aging activities, such as protection against aggregation-prone proteins and oxidative stress in *C. elegans* and promotion of the migratory capacity of human primary endothelial cells. These AhR-independent effects are largely mediated by the Nrf2/SKN-1 pathway.

## 1. Introduction

The aryl hydrocarbon receptor (AhR) is a ubiquitous ligand-activated transcription factor identified as a determinant for the toxicological response to 2,3,7,8-tetrachlorodibenzo-p-dioxin (TCDD) in mammals [1]. AhR signaling pathways have been well described in mammalian cells. Briefly, unliganded AhR is localized in the cytoplasm and stabilized by diverse co-factors, such as 90-kDa heat shock protein (Hsp90), the AHR-interacting protein (AIP), and the chaperone p23. Binding to exogenous (e.g., TCDD) or endogenous (e.g., kynurenine) ligands promotes the translocation of this complex into the nucleus, where AhR dissociates from its co-factors and assembles in a heterodimer with the AHR nuclear translocator (ARNT). The resulting AHR/ARNT complex binds to the xenobiotic responsive elements (XREs) of a battery of responsive phase I and II detoxification genes, eventually leading to the ligands’ degradation [2]. Apart from its role in xenobiotic response, functions for the AhR in a variety of pathophysiological processes, ranging from immunity [3,4], inflammation [5,6], lipid and glucose metabolism [7,8] to cardiovascular, liver and other organs’ diseases [9,10,11,12], have been discovered in the last decades. Growing evidence also points to disparate and seemly contradictory roles of AhR in the aging process, which could nonetheless be reconciled, taking into account tissue-, dose- and species-specific effects [13]. A negative role for AhR in aging and age-associated features has been described across species [14,15]. Compared to the wild-type *Caenorhabditis elegans,* the AhR mutant strain, *ahr-1(ju145)*, has an extended life- and health spans; in mice, AhR deficiency improves vessel function and increases activity of the nitric oxide synthase and, therefore, the NO bioavailability; and, finally, a positive correlation was found between AhR expression and vessel stiffness in middle-aged and aged human subjects [14]. Furthermore, in an epidemiological study on a Chinese population, AhR expression was related to the incidence of coronary arterial disease [16].

Of note, many compounds impacting aging or age-related diseases [17,18,19] can modulate AhR activity [20,21,22]. While the activation of AhR by xenobiotics leads to different cancers in mammals [23,24], dietary and environmental factors were shown to have opposite AhR-dependent effects on *C. elegans’* health span [25]. Among the dietary AhR modulators, polyphenols such as curcumin have been largely studied for their pro-health effects. Curcumin is a yellow pigment from *Curcuma longa*, with numerous evolutionarily conserved beneficial properties, including antioxidant, anti-inflammatory and anti-aging activities [26]. Curcumin prevents protein aggregation and increases longevity in *C. elegans* and *Drosophila* via modulation of protein homeostasis [27,28,29]. Moreover, when administered to an Alzheimer’s disease transgenic mouse model, it significantly reduced the total amyloid–beta (Aβ) burden [30]. In old mice, curcumin restored the NO bioavailability, thus reducing oxidative stress and improving endothelial dysfunction and artery stiffness assessed by aortic pulse wave velocity (PWV)—one of the most important clinical measurements or markers of large elastic artery stiffness [31]. Several studies in humans also showed a protective effect of curcumin on cardiovascular health [32]. However, the mode of action of curcumin is still largely unclear, and, more importantly, whether its beneficial health effects are mediated by AhR has not been investigated [33,34].

Model organisms, such as the nematode *C. elegans*, have been instrumental in identifying the genetic and environmental determinants of aging. This is due to its many advantageous properties, including its easy laboratory handling, short lifespan, and the production of a large number of progenies by self-fertilization. *C. elegans’* genome is completely sequenced, and most of its genes and pathways are evolutionarily conserved. The protein sequence of AhR is conserved during the evolution and in *C. elegans,* the orthologs of AhR and ARNT are encoded by the AhR-related (*ahr-1*) and *ahr-1* associated (*aha-1*) genes, respectively [35]. The corresponding proteins, AHR-1 and AHA-1, share about 40% of their sequence identity with the mammalian ones and form a heterodimer (also with the mammalian counterparts), which can bind the XREs of the target genes in vitro [36]. AHR-1 is mostly expressed in neuronal cell types, such as GABAergic neurons [37], and it plays a key role in controlling neuronal development [38]. Unlike vertebrates’ AhR, AHR-1 does not bind TCDD and other related xenobiotics [35,39], yet it shares with mammalian AhR common features in the regulation of neuronal processes, development and fertility [37,38,40,41,42,43], ultimately suggesting that the ancestral AhR was not directly involved in controlling genes for the degradation of toxic ligands [44,45]. We thus recognized *C. elegans* as a unique and powerful model system to identify and study the ancestral functions of the AhR possibly unrelated to its xenobiotics response.

In this study, we investigated the role of AhR in curcumin’s anti-aging effects across species. Through a combination of in vivo, in vitro, and in silico analyses, we found that curcumin displays different beneficial anti-aging effects through uncoupled *ahr-1*-dependent and -independent mechanisms. We found that *C. elegans ahr-1*-depleted animals are long-lived but more sensitive to oxidative stress. While curcumin did not further extend the lifespan of the *C. elegans ahr-1* mutants, it promoted their resistance to oxidative stress. Curcumin also promoted the antioxidant response and migratory capacity of human primary endothelial cells (EC) independently of AhR, an effect that primarily relied on Nrf2/SKN-1 across species. SKN-1 is the *C. elegans* ortholog of the human Nrf2 (nuclear factor erythroid 2-related factor 2) redox transcription factor, which plays an evolutionarily conserved role in cell and organismal homeostasis and response to oxidative stress [46,47]. Coupling results from a cellular reporter assay and in silico modeling of the AHR-1 ligand-binding domain (LBD), we then showed that curcumin most likely suppressed AHR-1 activity in a ligand-binding-independent manner. Notably, and in line with the data in *C. elegans* and EC, curcumin and pro-oxidants displayed opposite effects on AHR-1 activity, implying that curcumin may modulate AHR-1 activity through its antioxidant capacity either directly or indirectly via the regulation of Nrf2/SKN-1 or other redox regulatory proteins.

## 2. Materials and Methods

### 2.1. C. elegans

#### 2.1.1. *C. elegans* Strains and Cultivation

We used the following *C. elegans* strains: N2 [wild-type], CZ2485 [*ahr-1(ju145)*], NV38b [*ahr-1(ju145); unc-54p::Q40::YFP*], NV38wt [*unc-54p::Q40::YFP*] (original strain AM141 [48]), NV42a [*unc-54p::alphasynuclein::YFP, ahr-1(ju145)*], NV42wt [*unc-54p::alphasynuclein::YFP*] (original strain NL5901 [49]), NV35a [*ahr-1*(*ju145*); *(pAF15)gst-4p::GFP::NLS*], NV35wt [*(pAF15)gst-4p::GFP::NLS*] (original strain CL2166). For maintenance, worms were kept synchronized by egg lay at 20 °C on Nematode Growth Media (NGM) plates and fed with *E. coli* OP50, according to methods described in [25]. For the experiments, worms were synchronized on plates supplemented with *E. coli* HT115(DE3) on plates supplemented with 1 mM IPTG (Fisher Scientific, Geel, Belgium).

#### 2.1.2. Gene Silencing by RNA-Mediated Interference (RNAi)

Gene silencing was achieved through feeding *E. coli* HT115(DE3) expressing plasmids with dsRNA against specific genes [50]. RNAi feeding was applied continuously from birth to death. For juglone resistance assay with HT115(skn-1) bacteria, RNAi feeding was applied from L4 worms for 24 h before transferring them to fresh juglone plates.

#### 2.1.3. *E. coli* Strains and Growth

Bacteria were grown in LB medium at 37 °C overnight. When using *E. coli* carrying vectors, the LB medium was supplemented with 0.01% of ampicillin and 0.0005% of tetracycline. *E. coli* HT115(L4440), HT115(*ugt-45*), HT115(*skn-1*), and OP50 were obtained from the Ahringer *C. elegans* RNAi library [51].

#### 2.1.4. Lifespan

The lifespan analysis was started from a synchronized population of worms, which was transferred to fresh NGM plates daily during the fertile period. After the fertile phase, the animals were transferred every alternate day. Dead, alive, and censored animals were scored during the transferring process. Animals were counted as dead when they showed neither movement, nor response to a manual stimulus with a platinum wire, nor pharyngeal pumping activity. Animals with internal hatching (bags), an exploded vulva, or which died desiccated on the wall were censored. The number of dead and censored animals was used for survival analysis in OASIS [52] or OASIS 2 [53]. For the calculation of the mean lifespan and the survival curve in OASIS and OASIS 2, the Kaplan–Meier estimator was used, and the *p*-values were calculated using the log-rank test between pooled populations of animals.

#### 2.1.5. Movement/Health Span

The movement was set as a parameter for healthy aging, and the phase of the active movement is referred to as health span. It was assessed in the populations used for the lifespan assay. Animals, which were either crawling spontaneously or after a manual stimulus, were considered as moving, while dead animals or animals without crawling behavior were considered as not moving. Statistical analysis was done as described for lifespan.

#### 2.1.6. Curcumin Treatment of *C. elegans*

Curcumin (Sigma Aldrich, Taufkirchen, Germany) was dissolved in DMSO (Carl Roth, Karlsruhe, Germany) in a concentration of 100 mM and supplied to the NGM after autoclaving. The final concentration of curcumin in the media was 100 µM (0.1% DMSO). Control plates contained 0.1% DMSO. Worms were treated continuously starting from eggs.

#### 2.1.7. Quantification of PolyQ Aggregates

PolyQ_40_ aggregates were visualized by fluorescence microscopy (100× magnification) in 10-day-old worms anesthetized with 15 mM sodium azide (Sigma Aldrich, Taufkirchen, Germany). To assess the number of aggregates, images were stitched using the Fiji pairwise stitching plugin [54] to create whole worms, and the number of the aggregates was quantified in Fiji [55] using the plugin “Analyze Particles”.

#### 2.1.8. Quantification of α-Synuclein Aggregates

α-synuclein aggregates in the head muscles of 7-day-old worms were visualized by fluorescence microscopy (400× magnification) in worms anesthetized with 15 mM sodium azide (Sigma Aldrich, Taufkirchen, Germany, S2002). Pictures were segmented using Ilastik (version 1.3.0) (Laboratory of Anna Kreshuk at the European Molecular Biology Laboratory, Heidelberg, Germany) (available on https://www.ilastik.org/, accessed on 10 April 2018) [56]. The segmented pictures were used to analyze the number of aggregates in Fiji [55] using the plugin “Analyze Particles”.

#### 2.1.9. Microarray and GO Term Analysis

Samples from five independent replicates with approximately 1000 3-day-old worms per condition were collected, and the RNA was extracted and loaded to an Affymetrix Chip. The microarray raw data in the format of CEL were analyzed using the software R (version 3.4.2) (R Foundation, Vienna, Austria) and Bioconductor (The Bioconductor Project, Bioconductor is open source and open development) [57]. Background correction, normalization, and expression calculation were performed with the oligo package [58] and the RMA method. For quality control of the array, the package arrayQualityMetrics_3.34.0 [59] was used. Because of the quality measures, sample ahr-1C5 was excluded from further analysis. The differentially expressed genes were identified using the limma package and a linear model and moderated t-statistic with FDR to test for multiple comparisons [60]. A *p*-value of 0.1 was applied. The differentially expressed genes were analyzed for Gene Ontology term enrichment using Cytoscape (version 3.6.0) (The Cytoscape Consortium, an independent not-for-profit organization) [61] with the plugin ClueGo (version 2.5.0) (Laboratory of Integrative Cancer Immunology, Paris, France) [62]. The microarray data can be accessed through the Gene Expression Omnibus accession no. GSE195769.

#### 2.1.10. ROS Quantification

MtROS were detected in live wt and *ahr-1* mutants worms using MitoSOX Red (ThermoFisher Scientific, Dreieich, Germany). Nematodes were synchronized by egg laying onto IPTG plates using HT115(L4440) bacteria as food. Then, 48 h later, 50 animals at the L4 stage were transferred onto freshly prepared 10 µM MitoSOX Red plates seeded with UV-killed HT115(L4440). The worms were incubated in the dark at 20 °C. Following 16 h incubation, they were moved onto new NGM plates spread with live HT115(L4440) for 1 h to remove residual dye from the intestines. For imaging, nematodes were mounted onto 2% agarose pad slides, anesthetized by adding 10 mM levamisole and fixed by ProLong™ Glass Antifade Mountant (ThermoFisher Scientific, Dreieich, Germany). Images were acquired immediately with a Zeiss Axio Imager M1 microscope (Carl Zeiss, Inc., Cologne, Germany) using a 40× objective and a DsRed Filter. Afterward, the worm head region was manually selected, and the integrated intensity was calculated using the imaging software Fiji [55].

#### 2.1.11. Tetramethylrhodamine Ethyl Ester (TMRE) Assay

To assess the mitochondrial membrane potential, nematodes were synchronized by egg laying on IPTG plates seeded with HT115(L4440) bacteria. On the day of the experiment, TMRE (Invitrogen, Eugene, OR, USA) was dissolved in DMSO to a concentration of 5 mM and then diluted to 30 µM with heat-inactivated HT115(L4440) (30 min, 65 °C). A total of 150 µL of this solution was added per plate and left to dry in the dark for approximately 30 min. Sixty adult synchronous worms at 1, 3, or 5 days of adulthood were picked onto the TMRE plates prepared and left to the stain to absorb for 2 h in the dark at 20 °C. After staining, worms were transferred onto IPTG plates seeded with heat-inactivated HT115(L4440) and incubated for 1 h in the dark at 20 °C, to remove residual dye from the intestines. For imaging, 10 nematodes were mounted onto 2% agarose pad slides, anesthetized by adding 10 mM levamisole, and fixed by ProLong™ Glass Antifade Mountant (ThermoFisher Scientific, Dreieich, Germany). For each experimental run, 5 slides were prepared per group. Images were acquired immediately with a Zeiss Axio Imager M1 microscope (Carl Zeiss, Cologne, Germany) using a 2.5× objective and a DsRed filter. The fluorescence intensity was calculated using Fiji [55].

#### 2.1.12. Pharyngeal Pumping Rate and Motility Assay

N2 and CZ2485 nematodes were synchronized by bleaching [63] and the eggs were left to hatch in egg buffer (118 mM NaCl, 48 mM KCl, 2 mM CaCl_2_, 2 mM MgCl_2_, 25 mM Hepes, pH 7.3) overnight, on orbital shacking. L1 larvae were spotted onto NGM supplemented with 1 mM IPTG and containing 0.1% DMSO or 100 µM curcumin. HT115(L4440) bacteria were used as food. Young adult worms were collected with M9 buffer, centrifuged (300*g* × 3 min), and washed twice to remove bacteria. Worms were incubated with 0–1 mM H_2_O_2_ (Sigma-Aldrich, Taufkirchen, Germany) (100 worms/100 μL), for 2 h on orbital shaking. Control worms were incubated with M9 buffer only. After 2 h, worms were moved onto NGM supplemented with 1 mM IPTG and containing 0.1% DMSO or 100 µM curcumin and seeded with HT115(L4440) bacteria as food. The pharyngeal pumping rate, scored by counting the number of times the terminal bulb of the pharynx contracted over a 1 min interval (pumps/min), and the motility assay, scored by counting the number of body thrash (body bends/min) in M9 buffer over a 1 min interval, were scored from 2 up to 20 h later.

#### 2.1.13. Acute Juglone Sensitivity Assay

N2 and CZ2485 nematodes were synchronized by egg laying onto NGM plates with either DMSO or 100 µM of curcumin. Plates were supplemented with 1 mM IPTG and seeded with HT115(L4440) or HT115(*ugt-45*) bacteria as food. To evaluate the effect of *skn-1* RNAi, the worms were synchronized by egg laying onto DMSO or curcumin plates seeded with HT115(L4440) bacteria. As nematodes reached the L4 larval stage, they were transferred for 24 h onto DMSO or curcumin plates seeded with HT115(*skn-1*) bacteria. Day 1 adult worms (25 worms) were moved onto fresh NGM plates containing 200 µM Juglone (Merck, Darmstadt, Germany) and seeded with 25 µL of 10× concentrated bacteria overnight culture. Worm survival under juglone-induced oxidative stress was checked by touch-provoked movement hourly, for 6 h. Animals were scored as dead when they failed to respond to touch with a platinum wire pick. Nematodes desiccated on the wall were censored. The number of dead and censored animals was scored and the Online Application for Survival analysis OASIS 2 was employed for survival analysis [53].

#### 2.1.14. Quantification of the *gst-4*::GFP Intensity

NV35wt and NV35a were synchronized by egg laying onto NGM plates supplemented with 1 mM IPTG and containing 0.1% DMSO or 100 µM curcumin. HT115(L4440), HT115(*ugt-45*) and HT115(*skn-1*) bacteria were used as food. To visualize GFP fluorescence, day 1 adult worms were anesthetized with 10 mM levamisole hydrochloride solution and mounted on 2% agarose pads. Images were immediately acquired with a Zeiss Axio Imager M1 microscope (Carl Zeiss, Cologne, Germany), 2.5× magnification) and then analyzed using the software CellProfiler (The CellProfiler project team, Cimini Lab at the Broad Institute of MIT and Harvard, MA, USA). Briefly, images were processed using a pipeline to segment worms in each image from bright field microscopy and separate them from the background. Then, the integrated GFP intensity was measured per worm.

#### 2.1.15. Semi-Quantitative Real-Time PCR (qPCR) in *C. elegans*

Samples from 3 independent replicates with approximately 1000 3-day-old worms per condition were collected and RNA was extracted. After washing and elution, the RNA content was quantified by spectrophotometry, and 1–2 µg of RNA was used for the cDNA synthesis (Omniscript RT Kit (Qiagen, Hilden, Germany). Primer pairs are listed in Appendix A. For the Real-time qPCR, the cDNA was diluted at 1:20 in 10 mM TRIS (pH 8.0). For the reaction, the qPCR Green Core kit (Jena Biosciences, Jena, Germany)) or the GoTaq^®^ qPCR kit (Promega, Walldorf, Germany) was used. The samples were run in a MyiQ2 cycler (BioRad, Feldkirchen, Germany), and the expression of each sample was measured in duplicate on the same multi-well plate. The expression was calculated relative to the reference genes *act-1* and *cdc-42* using the iQ5 software. All data collected were enabled for gene study according to the BioRad user instructions, and the expression was calculated using the normalized expression (ddC_T_). The efficiency of each primer pair reaction was added for the correct quantification of the normalized expression. The efficiency was assessed with 1:20, 1:100, 1:500, and 1:2500 dilutions of the cDNA. From normalized expression values, the fold-change compared to wild-type was calculated for each replicate.

### 2.2. Mammalian Cells

#### 2.2.1. Cultivation of Cos7 Cells

Cos7 cells (ATTC, #CRL-1651) were cultivated at 37 °C and 5% CO_2_ in Dulbecco’s Modified Eagle’s Medium (DMEM) (Gibco/ThermoScientific, Dreieich, Germany) with 1% pyruvate, 1% Glutamax and 10% Fetal Bovine Serum (FBS) (Gibco/ThermoScientific, Dreieich, Germany) and additional Penicillin (10.000 units/mL)/Streptomycin (10.000 µg/mL). As soon as the cells built a confluent cell lawn, they were detached from the base by using 0.05% Trypsin/EDTA (Thermo Scientific, Dreieich, Germany).

#### 2.2.2. Transfection Plasmids

We used the following plasmids for the transfection of the Cos7 cells: pcDNA3, pcDNA3-AhR-1-VP16, pcDNA3-AhA-1-VP16, pcDNA3-AhR-1(ju145)-VP16, p1A1-FL, phRL-TK. The plasmids are described by Larigot et al. (submitted along with this study). The p1A1-FL plasmid carries an XRE-inducible luciferase, phRL-TK carries a renilla luciferase, and the pcDNA3-AhR-1-VP16 and pcDNA3-AhA-1-VP16 carry sequences for the expression of the *C. elegans* AHR-1 and AHA-1, respectively. For this study, we created pcDNA3-AhR-1(LBD)-VP16 by site-directed mutagenesis of the pcDNA3-AhR-1-VP16 plasmid with the QuikChange II Site-Directed Mutagenesis Kit (Agilent, Santa Clara, CA, USA). The following primer pair was used to create an L to A substitution at L363, and an H to Q substitution at H365 of AHR-1: LBD-F 5′-GAGAGCATCGGCGCGACCCAACGGCTGCTGAACGAG-3′ and LBD-R 5′-CTCGTTCAGCAGCCGTTGGGTCGCGCCGATGCTCTC-3′. Super-competent XL1-Blue cells were transformed with the obtained plasmid for amplification. The plasmid sequence was verified by Sanger sequencing.

#### 2.2.3. Transient Transfection of Cos7 Cells

At 24 h before transfection, 20,000 cells/well were seeded in a 48-well plate in 400 µL DMEM (+10% FBS + antibiotics) and incubated at 37 °C. Cells were then transfected with the following plasmid concentrations using lipofectamine 2000 (Invitrogen, Eugene, OR, USA): p1A1-FL (244 ng/well), phRL-TK (36 ng/well), pcDNA3-AhA-1-VP16 (5 ng/well), and either pcDNA3-VP16 (10 ng/well), pcDNA3-AhR-1-VP16 (5 ng/well) or pcDNA3-AhR-1*(ju145)*-VP16 (5 ng/well) as described by Larigot et al. (submitted along with this study). Lipofectamine 2000 was used in a concentration of 1 µL/well and pre-incubated with the respective plasmids in DMEM for 20 min before use. For the transient transfection, Cos7 cells were incubated with the lipofectamine/plasmid mix for 3 h in DMEM (+10% FBS) without antibiotics to avoid antibiotic-induced toxicity. Then, the transfection medium was removed and replaced by 400 µL/well DMEM (+10% FBS + antibiotics). The transfected cells were incubated at 37 °C. On each plate, 2 wells of cells were not transfected and thus used for normalization purposes.

#### 2.2.4. Treatment of Cos7 Cells

Stock solutions at concentrations 1000-times higher than the desired treatment concentration were prepared for all of the compounds. Curcumin (Sigma Aldrich, Taufkirchen, Germany), Benzo(a)pyrene (Sigma Aldrich, Taufkirchen, Germany), leflunomide (Sigma Aldrich, Taufkirchen, Germany), and lutein (Sigma Aldrich, Taufkirchen, Germany), were dissolved in DMSO (Carl Roth, Karlsruhe, Germany), while resveratrol (Sigma Aldrich, Taufkirchen, Germany) and rotenone (Sigma Aldrich, Taufkirchen, Germany) were dissolved in ethanol (Carl Roth, Karlsruhe, Germany). At 24 h after transfection, the cell culture medium of the cells was replaced with a cell culture medium containing a 1:1000 dilution of the respective compound. The cells were treated for 24 h before assessing the luciferase activity.

#### 2.2.5. Luciferase Assay (AhR Activity)

The AhR transcriptional activity was assessed by measuring the activity of an XRE-driven luciferase [64]. For this, a Dual-Luciferase Reporter Assay System (Promega, Walldorf, Germany) was used. After a 24 h treatment, Cos7 cells were washed twice with PBS and then lysed for 15 min at RT using the passive lysis buffer included in the Dual-Luciferase Reporter Assay kit. A total of 20 µL of the lysed cells were placed in a white 96-well plate and used for luminescence measurements. The luciferin (LARII) and renilla (Stop&Glo) substrates were prepared according to the manufacturer’s description. The samples were loaded on a luminometer (EG&G Berthold microplate Luminometer LB 96V Microluminomat plus (Berthold Technologies, Bad Wildbad, Germany) and the substrates were attached to the tubing system of the luminometer. First, 65 µL of LARII was added to each well of the sample, and the luminescence produced by the firefly luciferase was measured, then 65 µL of Stop&Glo reagent was added and the luminescence produced by the Renilla luciferase was measured. To assess AhR activity from the luminescence measurements, we performed the following post-processing steps: First, the luminescence of non-transfected cells was subtracted from the luminescence of each sample for background correction. In the next step, we normalized the luciferase luminescence to the renilla luminescence of the same sample to eliminate differences in the transfection rate and cell number. Another normalization step to the pcDNA-VP16 transfected cells was performed for each treatment group to remove AhR-independent effects on the XRE-driven luciferase.

#### 2.2.6. Cultivation of Primary Human EC

Human primary EC (Lonza, Cologne, Germany) was cultured in complete endothelial basal medium (EBM) (Lonza, Cologne, Germany) supplemented with 1 μg/mL hydrocortisone, 12 μg/mL bovine brain extract, 50 μg/mL gentamicin, 10 ng/mL human epidermal growth factor, and 10% (*v*/*v*) fetal calf serum at 37 °C and 5% CO_2_ until the third passage. After detachment with 0.05% (*v*/*v*) Trypsin/EDTA (Thermo Scientific, Schwerte, Germany), cells were cultured in 6 cm culture dishes or 6 well culture plates for at least 18 h before transfection or treatment.

#### 2.2.7. Transient Transfection of EC

Cells were transfected as described previously [65]. In brief, EC were transfected by using SuperFect^®^ Transfection Reagent (Qiagen, Hilden, Germany) according to the manufacturer’s instructions. The overexpression or knockdown of AhR was achieved after 24 or 48 h, respectively. The transfection efficiency upon overexpression was approximately 40%.

#### 2.2.8. Scratch Wound Assay of EC

For investigation of the migratory capacity of EC, scratch wound assays were performed as described previously [66]. In detail, wounds were set into a cell monolayer with a cell scraper along a trace line. After the injury, non-attached cells were removed by gentle washing. The curcumin treatment was performed directly after the wound was set. Curcumin was dissolved in DMSO and used at the final concentration of 7.5 µM. EC migration was quantified by staining the cells with 5 µg/mL 4′, 6-diamidino-2-phenylindole (DAPI) (Carl Roth, Karlsruhe, Germany) in PBS for 5 min after the cells had been fixed with 4% (*v*/*v*) paraformaldehyde for 15 min at room temperature. Images were taken using a Zeiss AxioVision Observer D1 fluorescent microscope (Carl Zeiss, Oberkochen, Germany)) using a 200-x magnification. Cells migrated into the wound from the trace line were automatically counted using the particle analysis function of ImageJ 1.52a [67] after overlapping nuclei were separated.

#### 2.2.9. Immunostaining of EC

EC were fixed with 4% (*v*/*v*) paraformaldehyde for 15 min at room temperature. After permeabilization and blocking in 0.3% (*v*/*v*) Triton-X 100 and 3% (*v*/*v*) normal goat serum in PBS, cells were incubated with a rabbit antibody against AhR (1:100, Abcam, Cambridge, United Kingdom) or Nrf2 (clone D1Z9C, 1:100, Cell Signaling Technology, Frankfurt, Germany) diluted in 1% (*v*/*v*) normal goat serum in PBS overnight at 4 °C. Then, cells were washed with PBS and incubated with an Alexa 594-coupled goat anti-rabbit IgG (1:500, Invitrogen, Darmstadt, Germany) for 1 h at room temperature. For actin staining, cells were incubated with Alexa Fluor™ 488 Phalloidin (1:70, Invitrogen, Darmstadt, Germany) for 20 min at room temperature. Nuclei were counterstained with 0.5 μg/mL DAPI in PBS for 5 min at room temperature and cells were mounted with ProLong™ Diamond Antifade Mountant (Invitrogen, Darmstadt, Germany). Fluorescent images were taken using a Zeiss AxioVision Observer D1 fluorescent microscope, 400× or 200× magnification.

#### 2.2.10. qPCR in Cells

The total cellular RNA was isolated by combining lysis in TRIzol™ with downstream processing using the RNeasy Mini Kit (Qiagen (Hilden, Germany)) according to the manufacturer’s instructions. cDNA synthesis was performed using the QuantiTect^®^ Reverse Transcription Kit (Qiagen (Hilden, Germany)) with 1 µg RNA according to the manufacturer´s instructions. Relative transcript levels were determined by pPCR using the 2x SYBR^®^ Green qPCR Master Mix and a Rotor-Gene^®^ Q thermal cycler (Qiagen (Hilden, Germany)). The transcript for the ribosomal protein L32 (*rpl32*) was used as a reference, and the relative expression was calculated by the ΔC_t_-method [68]. The following intron-spanning primer pairs were used: *cyp1a1* (gene accession number NM_000499.5): 5′-TCGCTACCTACCCAACCCTT-3′, 5′-TGTGTCAAACCCAGCTCCAA-3′; *ahr* (gene accession number NM_001621.5): 5′-CGTGGGTCAGATGCAGTACA-3′, 5′ ACCAGGGTCAAAATTGGGCT 3′; *sod2* (gene accession number NM_000636.4): 5′-GCCCTGGAACCTCACATCAA-3′; 5′-AGCAACTCCCCTTTGGGTTC-3′; *rpl32* (gene accession number NM_000994.4): 5′-GTGAAGCCCAAGATCGTCAA-3′, 5′-TTGTTGCACATCAGCAGCAC-3′.

### 2.3. Mice

#### 2.3.1. Mouse Lines and Breeding

Female 8–12-week-old (“young”) and 18-month-old (“old”) AHR-deficient B6.129-AHR^tm1Bra/J^ (Schmidt et al., 1996; referred to here as AHR-KO) mice were bred as heterozygotes in the IUF´s animal facility. Wild-type littermates were used for control. Mice were bred and kept under specific pathogen-free conditions on a 12/12 h light–dark cycle and received standard chow (ssniff^®^M-Z, SSNIFF, Soest, Germany) ad libitum.

#### 2.3.2. qPCR in Mice

Total RNA was isolated from organ tissues of three WT and three AHR deficient mice with TriZol^®^. Then 400 ng of RNA was reverse transcribed using the reverse transcriptase M-MLV (Promega, Madison, WI, USA) and random hexamer primers. Gene expression levels were measured in duplicate for each mouse tissue on a Rotor-Gene Q (Qiagen, Hilden, Germany), in 15 µL final volume, containing 7.5 µL Rotor Gene SybrGreen™ (Biorad, Feldkirchen, Germany), 1 µM of each primer, 1.5 µL cDNA and RNase free water. Primer efficiencies were between 90% and 146%. See Appendix A for primer sequences and efficiencies. Expression levels were calibrated to the expression of RPS6 as a house-keeping gene in the same sample using the 2^−ΔΔCT^ method [69].

### 2.4. In Silico Analyses

#### Homology Modeling of the CeAhR LBD

The structural model of *C. elegans* AhR LBD (residues 267–372) was generated by homology modeling. The X-ray structures of the PASB domains of homologous bHLH-PAS family members sharing the highest sequence identity (about 20%) with the CeAhR PASB were used as templates: the circadian locomotor output cycles kaput (CLOCK, PDB: 4F3L), the neuronal PAS domain-containing protein 3 (NPAS3, PDB: 5SY7), the Hypoxia-inducible factors 2α (HIF2α, PDB: 3H82, 4ZP4, 3F1N) and 1α (HIF1α, PDB: 4H6J). The model was obtained with MODELLER [70,71,72]. The optimal model was selected from among the 100 generated, based on the best DOPE SCORE [73]. The quality of the models was evaluated using PROCHECK [74]. Secondary structures were attributed by DSSPcont [75]. The binding cavity within the modeled LBDs was characterized using the CASTp server [76]. Visualization of the models was accomplished using PYMOL [77].

### 2.5. Statistical Analysis

Unless otherwise stated, statistical analyses were performed in GraphPad Prism (Version 6.01) (GraphPad Software, Inc. San Diego, CA, USA). For life/health span assays, statistical analysis was done using OASIS [53]. Statistical analysis of the microarray data was performed in R (R Foundation (Vienna, Austria)). Boxplots were created in GraphPad Prism (Version 6.01) (GraphPad Software, Inc. San Diego, CA, USA) and show the median (line), 25–75th percentile (box) and 10–90th percentile (whiskers).

## 3. Results

### 3.1. Curcumin Promotes Health Span in an AhR-Dependent and -Independent Manner

Loss of *ahr-1* promotes *C. elegans’* health- and lifespan in basal conditions [14] and it negatively impacts age-related traits in response to mammalian AhR modulators, such as benzo[a]pyrene (BaP), UVB light, and microbiota [25]. Dietary polyphenols, such as curcumin, form an important group of mammalian AhR modulators with pro-longevity effects in *C. elegans* [78,79] and we thus investigated the lifespan-extending effect of curcumin for its *ahr-1* dependency. Curcumin reproducibly and significantly extended the life- and health spans of *C. elegans* in an *ahr-1*-dependent manner (Figure 1A,B). Previously, we have shown that loss of *ahr-1* also extends the lifespan in Huntington’s disease and Parkinson’s disease models, with muscle-overexpression of aggregation-prone polyglutamine (polyQ40) and α-synuclein (α-syn), respectively, while at the same time increasing their content of protein aggregates [25]. Interestingly, curcumin treatment increased the number of polyQ40 and α-syn aggregates to the same extent as *ahr-1* loss of function (Figure 1C). Curcumin also promoted lifespan and locomotory ability in these disease models (Figure 1D,E), but the effects of *ahr-1* loss and curcumin supplementation were additive in the PolyQ background (Figure 1D), revealing AHR-1-independent protective functions of curcumin at least in this compromised background.

### 3.2. ugt-45 Mediates the Anti-Aging Effects of Curcumin and ahr-1 Depletion

In search of possible downstream *ahr-1*-dependent effectors of curcumin, we took targeted and unbiased approaches. We examined the expression of classical mammalian AhR target genes and focused on the *Cyp* genes since curcumin alters *Cyp1A1* and *Cyp1B1* expression in mammalian cells [80,81]. However, the quantification of 47 different *cyp*s in *C. elegans* by semi-quantitative real-time PCR (qPCR) revealed that only *cyp-13B1* was significantly up-regulated either by *ahr-1* depletion or by curcumin in an *ahr-1*-dependent manner (Appendix A), while the three other *cyps* (i.e., *cyp-13A5*, *cyp-13A8* and *cyp-42A1*) were increased by curcumin only in the absence of *ahr-1* (Appendix A). These data, along with other works [25,82], suggest that *cyps* are likely not the major targets of *CeAhR*. This is also supported by our transcriptomic analysis in wild-type and *ahr-1* mutants. Indeed, consistent with the role of AHR-1 in neuronal determination [37,38,40,41], the gene expression changes between wild-type and *ahr-1* mutants showed enrichment in processes linked to neuronal development and differentiation and no major changes in classical detoxification genes (Figure 2A). qPCR analysis of some of the most up- and down-regulated genes between *ahr-1(ju145)* and wild-type (*atf-2, K04H4.2, egl-46, T20F5.4, ptr-4, dyf-7, clec-209, C01B4.6, C01B4.7, F56A4.3*) mostly confirmed their *ahr-1*-dependency in basal conditions (Figure 2B) but neither UVB [25] nor curcumin (Figure 2C) significantly affected the expression of these genes. We wondered whether the expression changes in these genes are evolutionarily conserved and assessed their expression in different tissues (i.e., brain, liver, gut, and blood) of 8- and 18-month-old wild-type and AhR KO mice. Some genes showed a tendency for an increased expression in young (*atf-2* homolog) or old (*lpr-4/5* homologs) mice in a tissue-specific manner, but neither an obvious pattern nor conserved changes were observed (Appendix A). These results reflect possible species-specific differences or tissue-dependent AhR transcriptional activity in mammals overlooked by whole-animal transcriptomic analysis in *C. elegans*.

A thorough examination of the most differentially expressed genes between *C. elegans* wild-type and *ahr-1(ju145)* revealed that the expression of many of these genes is affected in *C. elegans* during aging and by dietary mammalian AhR modulators (e.g., quercetin and resveratrol) [25], thus suggesting a role for AHR-1 in polyphenol-modulated gene expression. In line with this scenario, the microarray data showed that most of the genes differentially expressed upon curcumin treatment were indeed regulated in an *ahr-1*-dependent manner (Figure 2D; Table 1). Out of 47 genes altered by curcumin in the wild-type (43 up- and 4 down-regulated), only 5 were also induced by curcumin in *ahr-1(ju145)*. Among the genes regulated by curcumin in an AHR-1-dependent manner were phase II enzymes and interestingly, some of them (*ugt-9 and ugt-29*), were regulated in the same direction by curcumin or by loss of *ahr-1* (Table 1). Thus, we checked their expression and that of additional *ugt*s (*ugt-45* and *ugt-57*) that were differentially expressed when applying a less restrictive statistical analysis not corrected for multiple comparisons. Of the examined genes, *ugt-45* was increased in *ahr-1(ju145)* and by curcumin treatment (Figure 3A,B). We also observed changes in the expression of some detoxification genes between wild-type and AhR KO mice in a tissue-dependent manner. The differential expression of those genes was highest in the brain, where *Ugt2a3* (*ugt-9* and *ugt-29* in *C. elegans*) was down- and *Hpgds* (*gst-4* in *C. elegans*) was up-regulated (Appendix A). There was no change in the expression of any of the tested genes in the liver samples of mice (Appendix A). In line with the *C. elegans* data, the *ugt-45* murine homolog *Ugt3a2* showed a tendency toward overexpression in the intestines of Ahr KO mice (Appendix A). Notably, *ugt-45* RNAi prevented the beneficial effects on life- and health spans promoted by curcumin (Figure 3C,D) or *ahr-1* depletion (Figure 3E,F) indicating that the two interventions may rely on jointly modulated downstream signaling to elicit their anti-aging activity.

### 3.3. AHR-1 and Curcumin Independently Protect against Oxidative Stress

The beneficial properties of polyphenols are often ascribed to their ability to protect against reactive oxygen species (ROS) [83,84]. Since AhR is involved in oxidative stress-mediated processes [85,86,87], we wondered whether curcumin may impact animals’ physiology via AhR-regulated antioxidant responses. We observed that *ahr-1* mutants produce more mitochondrial(mt)ROS and have a reduced mitochondrial membrane potential (Figure 4A,B), two parameters correlating with longevity [88,89]. While consistent with the mitohormesis paradigm *ahr-1(ju145)* produce slightly more mtROS and live longer, these animals were more sensitive to oxidative stress than the wild type. Specifically, the detrimental effects induced by juglone and H_2_O_2_ on the animals’ pumping, motility and survival were significantly stronger in *ahr-1**(ju145)* compared to the wild type (Figure 4C–F). These data suggest that AHR-1 depletion has beneficial mitohormetic effects in basal conditions, while its presence is required for oxidative stress protection, thus uncoupling two often correlating age-related parameters, namely lifespan and stress resistance. Instead, curcumin significantly improved H_2_O_2_ and juglone resistance in both wild-type and *ahr-1* mutants (Figure 4E,F), suggesting that curcumin elicits an *ahr-1*-independent antioxidant response. Consistent with the uncoupled regulation of lifespan and oxidative stress resistance, *ugt-45* silencing did not affect sensitivity to oxidative stress either for *ahr-1* mutants or curcumin-treated animals (Figure 4G). Thus, curcumin has pro-longevity effects via *ahr-1* and *ugt-45,* but protects against oxidative stress through *ahr-1*-independent mechanisms.

### 3.4. Nrf2/SKN-1 Mediates the AhR-Independent Effects of Curcumin

To further evaluate AhR–curcumin crosstalk in additional age-related features, we measured the migratory capacity in human primary EC—a hallmark for vessel functionality, which declines with age [90] and is reduced by AhR activation [14]. In line with the anti-aging activity of *ahr-1* suppression and of curcumin, AhR overexpression significantly inhibited, while curcumin increased, the migratory capacity of primary human EC (Figure 5A). Of note, the induction of migratory ability by curcumin was comparable in empty vector- or AhR expression vector-transfected cells (Figure 5A). However, the migration of curcumin-treated cells was significantly reduced by AhR overexpression: curcumin induces in empty vector-transfected cells up to 60 migrated cells per high power field, while in AhR overexpressing cells only up to 25 cells per high power field (Figure 5A). These data suggest that the pro-migratory effect of curcumin is modulated by AhR-independent mechanisms but possibly also by a reduction in AhR activity. We next determined intracellular AhR distribution and the expression of *cyp1a1* in curcumin-treated human EC. Curcumin did not affect AhR–nuclear translocation (Figure 5B) or *cyp1a1* expression (Figure 5C).

In search of pathways modulated by curcumin in an AhR-independent manner, we turned back to nematode transcriptomic profiles to find transcription factors regulating genes significantly modulated by loss of *ahr-1* or by curcumin treatment in wild-type animals (Table 1). This in silico search identified the redox transcription factor SKN-1, the ortholog of human Nrf2 (nuclear factor erythroid 2-related factor 2), whose activation by curcumin [91] is often reported as a possible mediator of its anti-oxidant activity [92,93]. Accordingly, the prototype *C. elegans* Nrf2/SKN-1-dependent gene, *gst-4*, is overexpressed in the *ahr-1* mutant [25] and induced by curcumin in wild-type and even more in *ahr-1* mutants (Figure 5D). Moreover, curcumin increased stabilization and nuclear translocation of Nrf2 in primary human EC (Figure 5E) and induced the expression of manganese superoxide dismutase (*Sod2)*—a classic Nrf2 target gene—in the cells transfected with an empty vector or in cells in which AhR is silenced by shRNA (Figure 5F). The lack of *Sod2* induction by AhR shRNA in EC may be due to a partial reduction (50%) in *AhR* expression (Appendix A), which may not be sufficient to trigger the activation of Nrf2 or of additional transcription factor (TF), which, in *C. elegans,* might concur to the induction of the *gst-4* [94] upon complete AhR depletion. Interestingly, *Hpgds*, a homolog of *C. elegans gst-4*, was significantly increased in the brain of AhR KO mice (Appendix A) but it is not a target of Nrf2. Further evidence for possible Nrf2/SKN-1 independent signaling activated by *ahr-1* depletion is that the activation of the *gst-4* by curcumin is completely suppressed by *skn-1* RNAi in the *C. elegans* wild type, whereas *ahr-1* mutants still induce *gst-4* despite *skn-1* depletion (Figure 5G,H). However, *skn-1* RNAi reduced oxidative stress resistance in both wild-type and *ahr-1**(ju145)* (Figure 5I). Unexpectedly, *skn-1* silencing did not affect the juglone resistance of curcumin-treated animals (Figure 5I). Our data reveal a complex scenario whereby curcumin promotes different anti-aging features relying either on AhR-dependent or AhR-independent but Nrf2/SKN-1-dependent (and additional) signaling.

### 3.5. Curcumin and Pro-Oxidants Display Opposite Effects on AHR-1 Activity

Consistent with the anti-aging effect of reduced AhR expression/activity, our data suggest that curcumin may extend the lifespan of *C. elegans* by suppressing AHR-1-regulated pathways through reduction of AHR-1 expression/activity or acting on common downstream signaling pathways. Hence, we tried to quantify AHR-1 activity in *C. elegans,* but numerous attempts to evaluate AHR-1 expression and subcellular localization using antibodies (against mammalian AhR or customized antibodies against CeAhR) or fluorescently tagged reporters (OP562, UL1709, ZG93) did not give meaningful evidence. Considering that AHR-1 binds to XREs in vitro [35], we thought to use XRE-driven gene expression as a readout for AHR-1 activity. Thus, we turned to monkey-derived Cos7 cells, which do not express endogenous AhR and thus display no endogenous AhR activity [95,96] and can be exploited to monitor XRE-driven Luciferase induction as a readout for AHR-1 activity (Larigot et al.; submitted along with this study). When Cos7 cells were co-transfected with vectors expressing *C. elegans* AhR/*ahr-1*, ARNT/*aha-1*, and a luciferase-coupled XRE-containing promoter of the human *CYP1A1* gene [64] AHR-1 showed low activity in basal (vehicle-treated) conditions. Of note, treatment with curcumin or other nutraceuticals that promote healthy aging in *C. elegans*, such as lutein [97] and resveratrol [98,99], significantly suppressed AHR-1 activity (Figure 6A–C). Instead, BaP and leflunomide, known AhR activators in mammals, did not affect AHR-1 activity (Figure 6A,B) at the concentrations we used. Notably, AHR-1 activity was abolished in Cos7 cells transfected with a vector expressing the *ahr-1(ju145)* allele instead of the wild-type allele (Figure 6A–C), suggesting that *ju145* is a true loss-of-function allele and that the measured luciferase intensity is due to functional AHR-1.

We then sought to investigate whether curcumin reduces the activity of AHR-1 by direct binding or indirect modulation. To date, no ligands of *C. elegans* AHR-1 have been identified, and since there is no available information on its LBD, we performed an in silico analysis to characterize it. The two AHR-1 isoforms, 1a and 1b, were aligned and although different in length, their PASB domain sequence is identical. This sequence was then aligned to the PASB domain of *Drosophila melanogaster*, and to those of two AhRs from vertebrates for which structural models were previously generated, namely mouse (*Mus musculus*) [100], and zebrafish (*Danio rerio*) [101] (Figure 6D). The alignment showed clear differences between species with the main peculiarity of invertebrates baring sequence deletions in the most variable region in the PAS domain, corresponding to the flexible region, including the helical bundle (Cα, Dα, Eα helices) and the short loops connecting these elements (Figure 6D,E). These deletions could reduce the available space in the binding cavity of these AhRs. We then generated a 3D model of the AHR-1 PASB by homology modeling. This model presents the typical PAS fold, but with a shorter Dα helix compared to other AhRs. However, the internal cavity has some peculiarities; it contains more hydrophobic residues and is truncated in half by some internal side chains. In particular, H365 and H274 are faced and could form a hydrogen bond in the middle of the cavity; moreover, the Y332, L363, and L302 side chains could obstruct the cavity, reducing the internal space available for ligands (Figure 6E). This small and truncated cavity most likely does not allow the binding of large ligands (e.g., TCDD or curcumin). Similar to the AHR-1 structural model, a model of the zebrafish zfAhR1a showed that the LBD cavity is truncated compared to the TCDD-binding paralogs zfAhR1b and zfAhR2 [101]. Small and flexible ligands, such as leflunomide, bind and activate the zfAhR1a, but the leflunomide concentration we tested did not activate AHR-1 in our Cos7 cell system (Figure 6B).

We then wondered whether mutations in amino acids responsible for the small cavity of the LBD might allow classical ligands to activate CeAhR. The CeAhR L363 residue (Figure 6D) corresponds to A375 in mAhR^b−1^ and V375 in mAhR^d^, and this residue has a major impact on ligand binding [102]. Similarly, T386 of zfAhR1a (Figure 6D), matching to A375 in mAhR^b−1^ and A386 in zfAhR1b and zfAhR2, contributes to the lack of TCDD binding of zfAhR1a and, when mutated to alanine, restores TCDD sensitivity when Y296H is also introduced [101]. The amino acid Y296 is already a histidine in *C. elegans* (H274). Thus, we mutated only the leucine at the position L363 in the CeAhR vector to an alanine (L363A) (Figure 6D,E indicated by an arrow). Moreover, we mutated the nearby histidine at position H365 to glutamine (H365Q), which is Q377 in mice (Figure 6D,E indicated by an arrow) since it likely forms a hydrogen bond with H274 and might contribute to the small cavity of AHR-1 (Figure 6E). We then tested whether mammalian AhR ligands affect the AHR-1 activity when L363 and H365 are mutated. However, these alterations, instead of restoring response to xenobiotic ligands as in zebrafish [101], abolished even the basal AHR-1 activity, similar to the *ju145* allele (Figure 6F). These results show clear differences between the LBDs of *C. elegans* and zebrafish but display that the LBD is fundamental for basal AHR-1 activity. Together with previous studies [35,38,82], our results suggest that AHR-1 is unlikely to be involved in the classical xenobiotic-induced transactivation response, which thus may not be relevant to *ahr-1*-regulated physiological aging. Instead, plant-derived compounds might exert conserved effects at least in part via the suppression of AHR-1-modulated pathways. Our 3D model suggests that curcumin does not modulate AHR-1 activity by binding its LBD. Thus, the suppression of AHR-1 activity by curcumin could be due to its antioxidant effect. Consistent with this possibility, and the increased sensitivity of the *C. elegans ahr-1* mutants to oxidative stress, we found that AHR-1 activity is indeed increased by ROS-inducing agents. Namely, Cos7 cells treated with the pro-oxidant rotenone displayed increased AhR activity when transfected with either *C. elegans* or murine AhR but not when transfected with the *ju145* allele or the allele with LBD mutations (Figure 6G,H).

Overall, while CeAhR activation protects against oxidative stress early in life, its decreased expression counteracts aging and mediates the beneficial anti-aging effect of curcumin (Figure 7). Curcumin may thus help balance redox TF activation in a context- and time-dependent manner and favor AhR suppression directly through its antioxidant effect and/or through the activation of Nrf2/SKN-1 (or other TF), which may concurrently mediate the anti-aging activity of curcumin.

## 4. Discussion

AhR was originally discovered in mammals for its xenobiotic response activity induced upon binding of environmental toxicants or endogenous ligands, but modulators not relying on ligand binding also exist but are much less investigated. *C. elegans* represents a unique model organism to investigate AhR activities independent of its classical xenobiotic response since CeAhR does not bind prototype AhR ligands [35,39]. Using this model, we identified an evolutionarily conserved function for AhR in the aging process [14] and showed that some of the mammalian AhR modulators (i.e., bacteria, BaP, and UVB) affect aging parameters through AHR-1 in a context-dependent manner [25]. Here, we followed up on our previous findings with a more mechanistic investigation of AhR-regulated aging features across species by the dietary polyphenol curcumin. Our combined in vivo, in vitro and in silico analyses revealed a new and complex scenario: while curcumin promotes anti-aging features in nematodes and human primary EC at least in part in an AhR-dependent manner, its antioxidant effects in both species rely on AhR-independent but primarily Nrf2/SKN-1 dependent mechanisms.

We showed for the first time that curcumin delayed *C. elegans’* physiological aging in an AHR-1-dependent manner. In search of possible downstream *ahr-1*-dependent effectors of curcumin, we employed targeted and unbiased approaches and found that the majority of differentially regulated genes upon curcumin treatment are regulated in an AHR-1-dependent manner. Moreover, many of these genes displayed a similar expression pattern in AHR-1-depleted and curcumin-treated animals, suggesting that curcumin is promoting lifespan extension via suppression of AHR-1 activity. Surprisingly, neither the targeted nor the transcriptomic analysis indicated a major role for classical AhR targets gens such as *cyps*, which instead were found largely to be under-expressed in neurons (Larigot et al.; submitted along with this study). Interestingly, these findings may indicate whole animal transcriptomics may mask the neuronal-specific effects of AhR, in this specific case through *cyps* genes. We found that among the differentially expressed genes, many belong to phase-II-detoxification enzymes, such as *ugt-45*, which was increased by both *ahr-1* depletion (and in the brain of AhR KO mice) and curcumin treatment and to mediate their lifespan extension. Instead, curcumin treatment and *ahr-1* depletion increased the expression of another phase-II-detoxification enzyme, *gst-4*, through different mechanisms: the former relaying on, while the latter mainly independent of, Nrf2/SKN-1, a classical redox TF inducing *gst-4* upon oxidative stress in *C. elegans* [103]. Moreover, while curcumin induces Nrf2/SKN-1-dependent responses in *C. elegans* (*gst-4* expression) and human primary EC (*Sod2* expression and migratory capacity), it also protects *C. elegans* against oxidative stress in a SKN-1-independent manner. In *C. elegans,* curcumin cannot extend lifespan in the very sick *skn-1(zu67)* mutants [104], while *gst-4* can be induced in a SKN-1-independent manner by EGF signaling [94] and a crosstalk between the EGF pathway and the AhR was reported in mammals [105].

Interestingly, and opposite to the wild-type strain, we observed an AHR-1-independent effect of curcumin on health span in nematode models for Huntington’s and Parkinson’s disease, respectively. In these strains, curcumin treatment increased the number of protein aggregates to the same extent as AHR-1 deficiency, indicating either a protective effect of the protein aggregation itself and/or curcumin activation of pathways protecting against protein aggregation independently of *ahr-1* depletion. The influence of curcumin on protein aggregation is controversial: it was shown to inhibit fibril formation but also to bind pre-fibrillar/oligomeric species of amyloidogenic proteins, thereby accelerating their aggregation and reducing the overall neurotoxicity [106]. Of note, caffeine, which also protects against features of cardiovascular aging [66,107], also prevents Aβ-induced paralysis without decreasing Aβ aggregates but through the activation of the protective Nrf2/SKN-1-dependent pathway [108]. It will be important to assess whether the protective effect induced by curcumin or *ahr-1* depletion in the *C. elegans* disease models is mediated by mechanisms promoting the removal of oligomeric/pre-fibrillar species into less toxic aggregates and/or by the activation of other mechanisms, such as Nrf2/SKN-1, which may concurrently protect against proteotoxicity. Of note, detoxification enzymes may contain both XRE and ARE (antioxidant responsive elements), and an interplay between Nrf2/ARE and AhR/XRE regulated signaling has been described [109]. It will be thus interesting to clarify how curcumin promotes its different beneficial anti-aging effects through the balance between Nrf2 and AhR-regulated signaling.

Our combined approaches showed that curcumin inhibits AHR-1 activity. In mammals, it was suggested the AhR inhibitory effect of curcumin is mediated by direct LBD binding [110] or inhibition of the protein kinase C that phosphorylates AhR [79]. Another study indicated that the transcriptional activity of the AhR is dependent on the cellular redox status and the chromatin structure, which are both influenced by curcumin [111]. While AHR-1 does not bind TCDD, it binds XRE in vitro [36] but systematic studies, addressing the potential of polyaromatic hydrocarbons, or other mammalian AhR ligands to modulate AHR-1, are missing primarily due to the lack of suitable tools to assess that. Our studies attempt to fill this gap and, exploiting Cos7 cells expressing AHR-1 coupled to luciferase assays (Larigot et al.; submitted along with this study) and in silico modeling of *C. elegans* LBD, revealed that curcumin suppresses AHR-1 activity, but likely not by direct LBD binding. The in vitro assay used in our study confirmed that CeAhR is not activated through the classical xenobiotics signaling. Yet, it would not reveal activities due to AhR binding to DNA sequences other than the “classical” XRE found in *CYP1A1* such as the polyphenol(quercetin)-responsive XRE found in PON1 [112,113]. Immunostaining in human primary EC also argues against curcumin inducing AhR nuclear translocation, which, along with the promoting effect of the migratory capacity in AhR overexpressing cells, may also indicate that curcumin suppresses AhR activity.

We propose that the inhibitory effect of curcumin, rather than relying on AhR binding, involves its antioxidant ability, which may indeed be associated with, or even depend on, the activation of Nrf2/SKN-1. Mammalian AhR is activated by ROS via LBD-independent oxidative modification [85], but our data show that the induction of AHR-1 activity by the pro-oxidant rotenone requires the LBD. An indirect mechanism of ROS-mediated AhR activation is the formation of the potent AhR ligand FICZ [114]. Yet, FICZ is a big planar molecule that, according to our in silico model, would not fit the AHR-1 LBD. While the exact mechanism by which AHR-1 activity is promoted by ROS and inhibited by curcumin (either via direct ROS quenching or indirectly via activation of Nrf2 or other antioxidants regulatory genes) remains to be established, this is strongly supported by our findings: AHR-1 is activated by rotenone, and *ahr-1* mutants display more mtROS, reduced mitochondrial membrane potential and are more sensitive to H_2_O_2_ and juglone as well as to UVB and BaP [25], both of which produce ROS [115,116]. In this context, it is interesting to note that *ahr-1* mutants display mild alteration of mitochondrial functions, which resemble those of mitohormesis [117]. This suggests that *ahr-1* depletion (and possibly curcumin by inhibiting AHR-1) may promote health span through mild mitochondrial stress, which is known to extend *C. elegans* lifespan through detoxification genes similarly modulated in the *ahr-1* mutants [118,119]. Moreover, whether Nrf2/SKN-1 and mitochondria play a role in modulating AHR-1 activity upon curcumin treatment is an interesting possibility that remains to be validated.

Overall, taking advantage of the multiple features offered by the nematode *C. elegans* for in vivo studies, we suggest that the ancestral function of the AhR might be in the regulation of phase-II-enzymes related to antioxidant rather than xenobiotic responses. Opposite to the detrimental effects induced by high levels of ROS, the beneficial effects promoted by AhR-deficiency may be mediated by mild mitochondria stress and/or mild ROS production (mitohormesis), which also rely on Nrf2/SKN-1. We also provide strong evidence for the interaction between curcumin and the AhR. Curcumin inhibition of AhR signaling is evolutionarily conserved and is likely not mediated by binding to the AhR LBD, but rather through curcumin’s ROS-scavenging properties or the activation of Nrf2/SKN-1. The Nrf2 signaling pathway can indeed be activated by curcumin in different ways [91]. Finally, our data showed that curcumin promotes anti-aging effects also in an AhR-independent manner in both *C. elegans* (increased *gst-4* expression and oxidative stress resistance) and human primary EC (increased Sod2 expression and migratory capacity), which could also explain the additive effects of curcumin and loss of AHR-1 function on the health span of polyQ-expressing animals.

## 5. Conclusions

In conclusion, through an original combination of in silico, vitro, and in vivo approaches, we showed that while CeAhR activation protects against oxidative stress early in life, its decreased expression counteracts aging and mediates the beneficial anti-aging effect of curcumin (Figure 7). Curcumin may thus help in balancing the activity of different transcription factors involved in detoxification/antioxidant responses (suppress AhR and activate Nrf2) in conditions where these are altered (increase AhR and decrease Nrf2/SKN-1), such as aging or age-associated disorders. Our work adds a further level of complexity to the already vast multifunctional and context-specific activities of AhR, with important repercussion on organismal health and lifespan. Moreover, it opens the door to additional studies, using the nematode system to discover and investigate ancestral functions of the AhR, which are less likely to be identified in mammals.

## Figures and Tables

**Figure 1 antioxidants-11-00613-f001:**
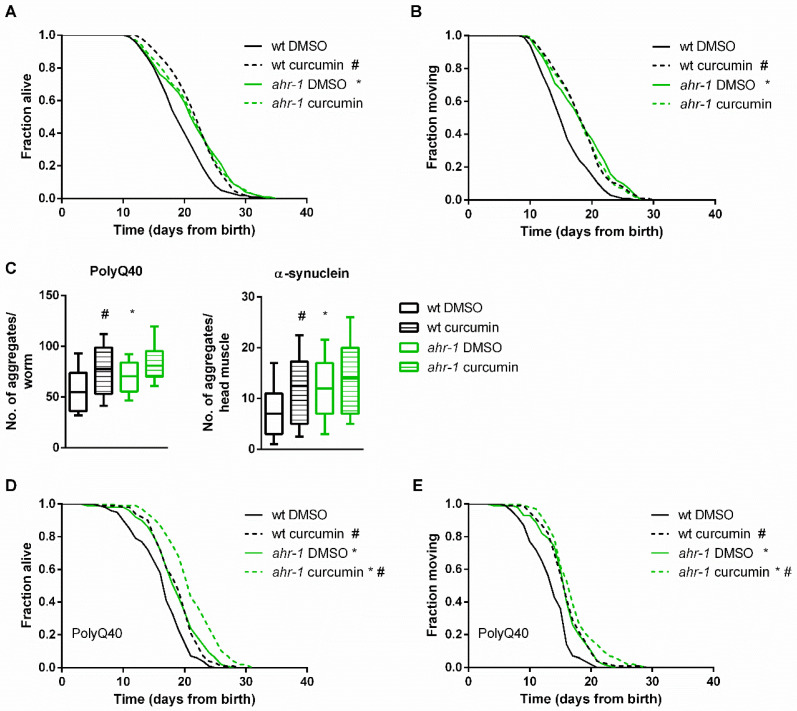
Curcumin promotes health in an AHR-1-dependent and -independent manner. Lifespan (**A**) and health span (**B**) curves of DMSO- or curcumin-treated wt and *ahr-1* nematodes. Survival curves show pooled data of 290–300 worms/condition in 5 experiments. Statistical test: log-rank test, ^#^ significance vs. DMSO, * significance vs. wt, Bonferroni *p*-value < 0.05. (**C**) Quantification of aggregates in 10-day-old polyQ;wt and polyQ;ahr-1 (left panel) or 7-day-old asyn;wt and asyn;ahr-1 (right panel). Boxplots show pooled data from 59–111 worms/condition in 3 experiments. Statistical test: 1-way ANOVA with Tukey’s multiple comparisons test, * *p*-value < 0.05 vs. wt, ^#^
*p*-value < 0.05 vs. DMSO. (**D**,**E**) Life/health span of polyQ;wt and polyQ;ahr-1. Survival curves show pooled data of 180 worms/condition in 3 experiments. Statistical test: log-rank test, ^#^ significance vs. DMSO, * significance vs. wt, Bonferroni *p*-value < 0.05.

**Figure 2 antioxidants-11-00613-f002:**
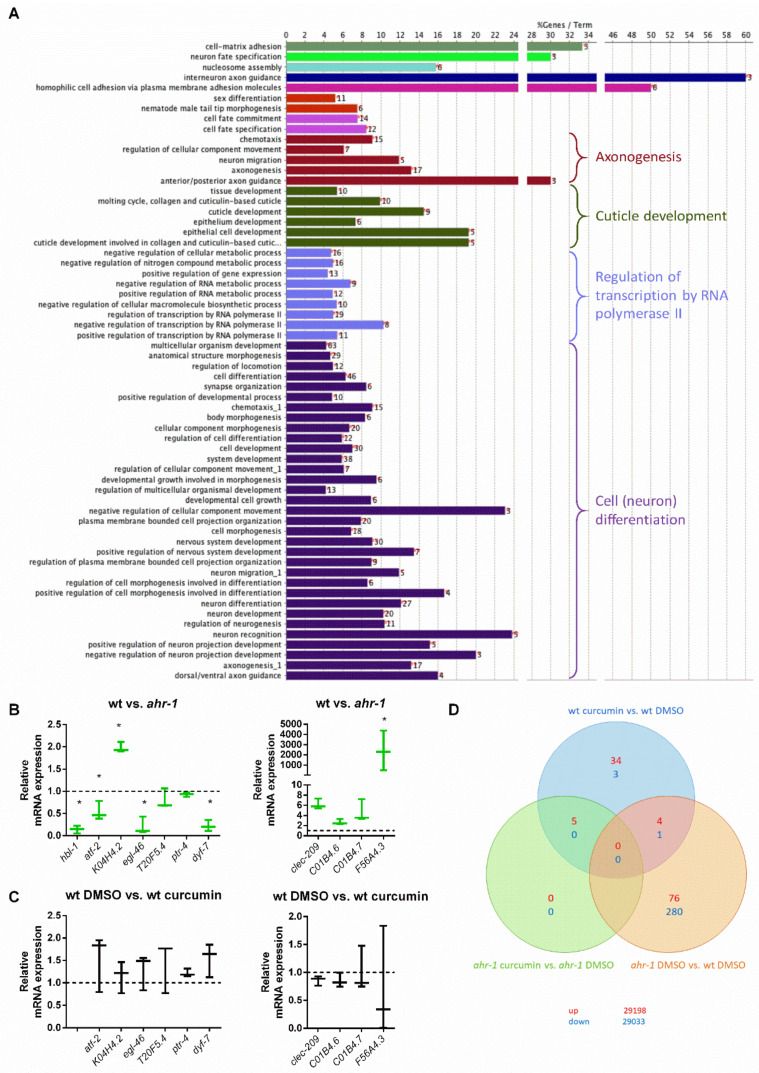
Genes differentially regulated by curcumin are primarily regulated in an *ahr-1*-dependent manner. (**A**) Gene ontology (GO) enrichment for biological processes after GO term fusion in *ahr-1* vs. wt. (**B**,**C**) The expression of the strongest down- and up-regulated genes between wt and *ahr-1* [25] was assessed by qPCR in wt vs. *ahr-1* (**B**) and DMSO- vs. curcumin-treated nematodes (**C**). Boxplots show data of 3 experiments. The expression is shown relative to DMSO-treated wt (dashed line). Statistical test: 1-way ANOVA with Tukey’s multiple comparisons test, * *p*-value < 0.05 vs. wt. (**D**) Venn diagram of differentially expressed genes on the microarray. The number of genes that were differentially up- or down-regulated between the indicated conditions is shown in red and blue, respectively. The numbers in the interchanges refer to the genes that occurred in both comparisons. The values in the lower right corner show the number of genes on the array that were not differentially expressed.

**Figure 3 antioxidants-11-00613-f003:**
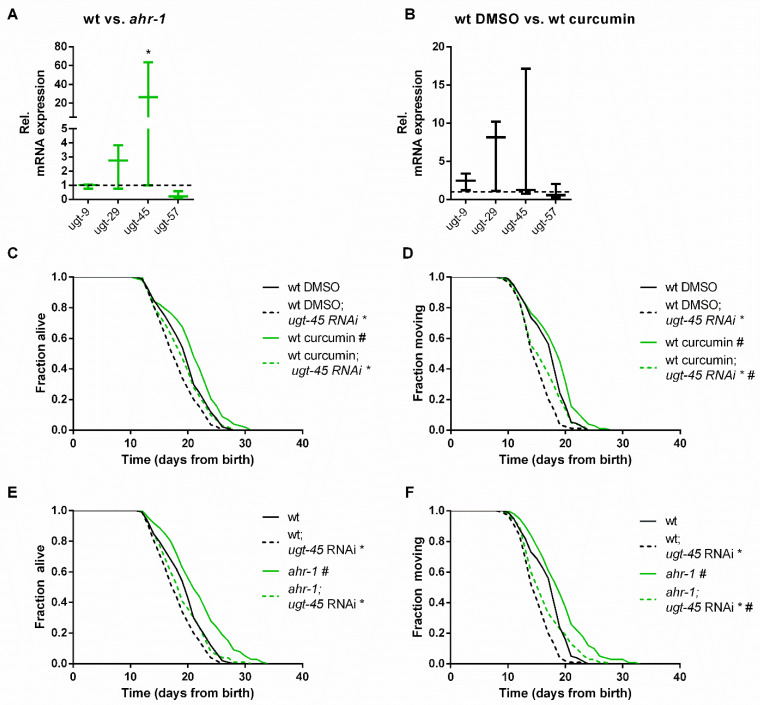
*ugt-45* is required for the lifespan extension of curcumin and *ahr-1* mutants. (**A**,**B**) Gene expression was assessed by qPCR in wt vs. *ahr-1* (**A**) and DMSO- vs. curcumin-treated wt nematodes (**B**). Boxplots show data of 3 experiments. The expression is shown relative to DMSO-treated wt (indicated as dashed line). Statistical test: 2-Way ANOVA with Sidak’s multiple comparisons test, * *p*-value < 0.05 vs. wt, ^#^
*p*-value < 0.05 vs. DMSO. (**C**,**D**) Effect of *ugt-45* RNAi on the curcumin-mediated life/health span extension in the wt. Survival curves show pooled data of 120 worms/condition in 2 replicates. Statistical test: Log-Rank test, ^#^ significance vs. DMSO, * significance vs. control RNAi, Bonferroni *p*-value < 0.05. (**E**,**F**) Effect of *ugt-45* RNAi on *ahr-1*-mediated life/health span extension. Survival curves show pooled data of 120 worms/condition in 2 replicates. Statistical test: Log-Rank test, ^#^ significance vs. wt, * significance vs. control RNAi, Bonferroni *p*-value < 0.05.

**Figure 4 antioxidants-11-00613-f004:**
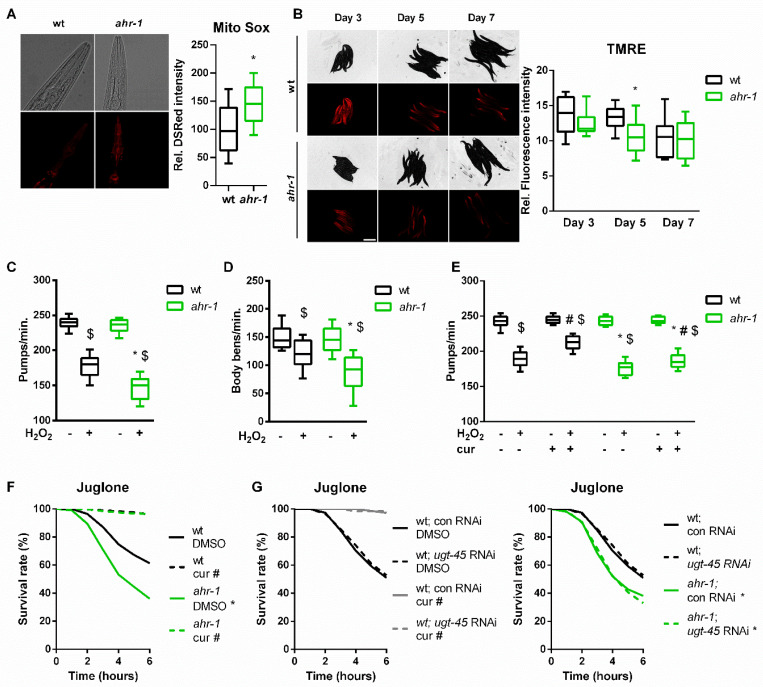
AHR-1 and curcumin independently protect against oxidative stress. (**A**) Representative images (left) and DSRed intensity quantification (right) in MitoSOX-stained wt or *ahr-1* nematodes. Boxplots show pooled data from 129–135 worms/condition in 3 experiments. (**B**) The mitochondria membrane potential was assessed by TRME staining in nematodes of indicated ages. Representative images (left) and the quantification of the TMRE fluorescence (right) are presented. Boxplots show pooled data from 3 experiments. (**C**,**D**) Pharyngeal pumping activity (**C**) and motility (**D**) of wt and *ahr-1* mutants after H_2_O_2_ treatment. Boxplots show pooled data from 39–54 (**C**) or 35–36 worms/condition (**D**) in 3–4 experiments. * *p*-value < 0.05 vs. wt, ^$^
*p*-value < 0.05 vs. control treatment, statistical test: 1-way ANOVA with Tukey’s multiple comparisons test. (**E**) Pharyngeal pumping of curcumin-treated nematodes after H_2_O_2_ treatment. Boxplots show pooled data from 32 worms/condition in 2 experiments. * *p*-value < 0.05 vs. wt, ^#^
*p*-value < 0.05 cur vs. DMSO treatment, ^$^
*p*-value < 0.05 H_2_O_2_ vs. control statistical test: 2-way ANOVA with Tukey’s multiple comparisons test. (**F**) Influence of curcumin on juglone-induced toxicity. Survival curves show pooled data of 500 worms/condition in 20 experiments. * significance *ahr-1* vs. wt, ^#^ significance curcumin vs. DMSO, Bonferroni *p*-value < 0.05. (**G**) Effect of *ugt-45* RNAi in curcumin-fed wt and *ahr-1* worms. Survival curves show pooled data of 150 worms/condition in 6 experiments. Statistical test: log-rank test, * significance *ahr-1* vs. wt, ^#^ significance curcumin vs. DMSO, Bonferroni *p*-value < 0.05. No statistical significance was observed in *ugt-45* vs. control RNAi-treated worms.

**Figure 5 antioxidants-11-00613-f005:**
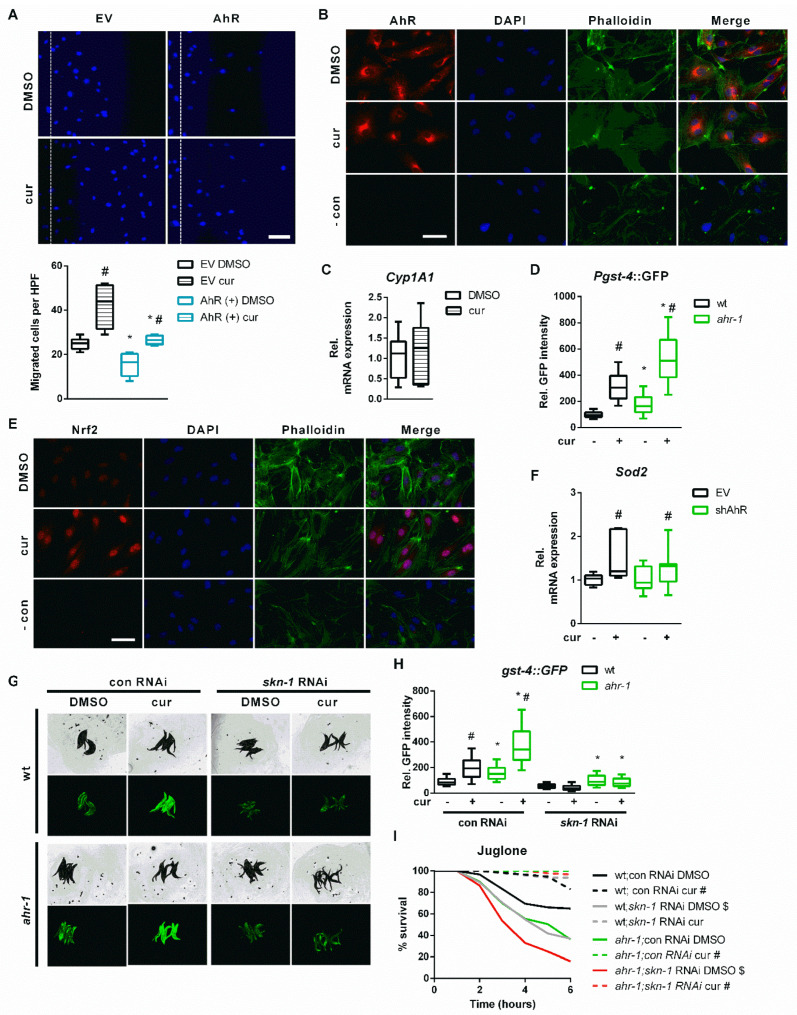
Curcumin activates Nrf2/SKN-1 independent of the AhR. (**A**) Scratch wound assay in curcumin (cur)- or DMSO-treated human primary EC transfected with an empty vector (EV) or an expression vector for human AhR. Upper panel: representative pictures; the dashed line represents migration start. Scale bar: 100 µm. Lower panel: quantification; boxplots show data of 4–6 experiments. Statistical test: 1-way ANOVA, * *p* < 0.05 vs. EV, ^#^
*p* < 0.05 vs. DMSO. (**B**,**C**) Human primary EC were treated with cur or DMSO. (**B**) Representative immunostainings: AhR is stained in red, nuclei were visualized with DAPI (blue), the cytoskeleton is counterstained with phalloidin (green), merge shows an overlay of all fluorescence channels. In the negative control (- con), the first antibody was omitted, and cells were stained with Alexa 488-coupled phalloidin and DAPI. Scale bar: 50 µm. (**C**) Relative *cyp1a1* expression was assessed by qPCR. Mean expression in the DMSO-treated controls was set to 1. Boxplots show data of 7 experiments. (**D**) *Pgst-4*:GFP expression in DMSO- and curcumin-treated (cur) wt and *ahr-1* worms. Boxplots show pooled data of 118–138 worms/condition in 4 experiments. * *p*-value < 0.05 vs. wt, ^#^
*p*-value < 0.05 vs. DMSO treatment, statistical test: 1-way ANOVA. (**E**) Representative immunostaining images of human primary EC treated with cur or DMSO: Nrf2 is stained in red, nuclei were visualized with DAPI (blue), the cytoskeleton is counterstained with phalloidin (green), merge shows an overlay of all fluorescence channels. In the negative control (- con) the first antibody was omitted, and cells were stained with Alexa 488-coupled phalloidin and DAPI. Scale bar: 50 µm. (**F**) Human primary EC were transfected with an empty vector (EV) or an expression vector for an shRNA targeting the human AhR transcript (shAhR). Relative *Sod2* expression was assessed by qPCR, mean expression in the EV transfected cells was set to 1. Boxplots show data of 7 experiments. ^#^
*p* < 0.05 vs. respective control. (**G**,**H**) *Pgst-4*::GFP expression in DMSO- or cur-treated wt and *ahr-1* nematodes subjected to control or *skn-1* RNAi. Representative images (**G**) and *gst-4*-driven GFP quantification (**H**) are shown. Boxplots show pooled data of 103–189 worms/condition in 4 experiments. (**I**) Juglone stress survival in curcumin- or DMSO-treated wt and *ahr-1* nematodes subjected to control or *skn-1* RNAi. Kaplan Meier survival curves show pooled data of 100 worms/condition in 4 experiments. Statistical test: log-rank test, * significance *ahr-1* vs. wt, ^#^ significance curcumin vs. DMSO, ^$^ significance *skn-1* vs. con RNAi, Bonferroni *p*-value < 0.05.

**Figure 6 antioxidants-11-00613-f006:**
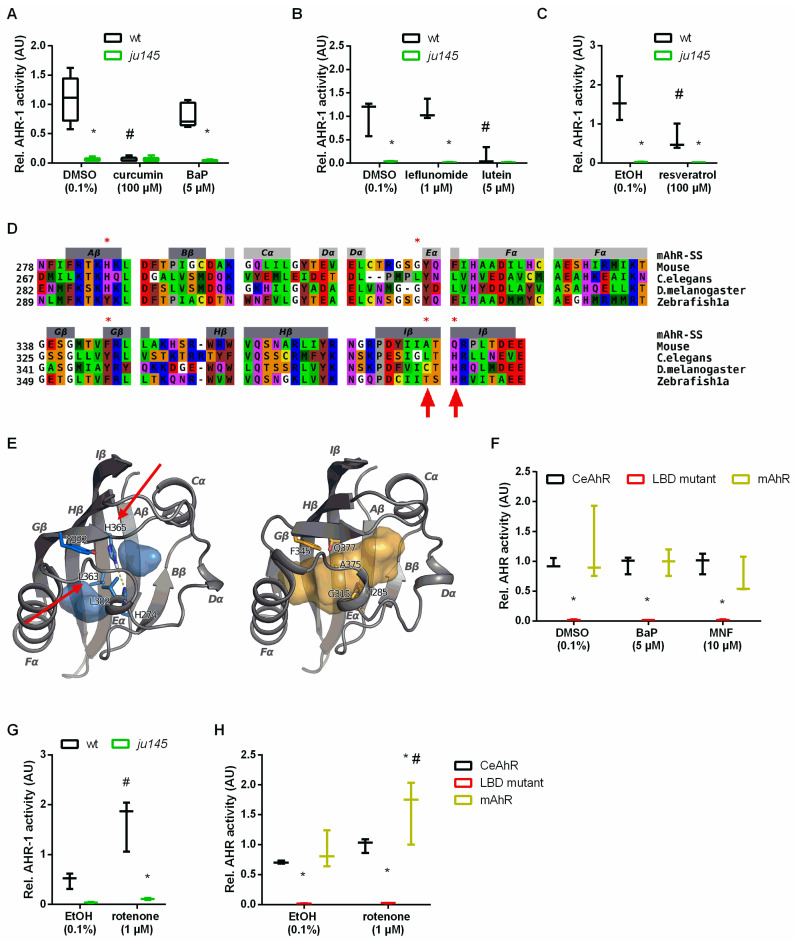
Curcumin and pro-oxidants have opposite effects on AHR-1 activity. (**A**–**C**) Evaluation of AHR-1 activity after treatment with the indicated compounds in Cos7 cells transfected with either wt AHR-1 (wt) or AHR-1 carrying the *ju145* point mutation (*ju145*) and AHA-1 as well as an XRE-inducible luciferase. Boxplots show data of 3–5 experiments. * *p*-value < 0.05 vs. wt, ^#^
*p*-value < 0.05 vs. DMSO/EtOH, statistical test: 2-way ANOVA and Tukey’s multiple comparisons test. (**D**) Alignment of the LBDs from *C. elegans*, *Drosophila*, and zebrafish AhRs. The color scheme for residues: red, acidic; blue, basic; purple, polar; yellow, Cys; brown, aromatic; green, hydrophobic; orange, Ser, Thr; gray, Pro; white, Gly. (**E**) Secondary structures attributed by DSSPcont to the CeAhR PASB are indicated on top (light gray bars for helices and dark gray bars for β-strands) and labeled according to the PAS domain nomenclature. Asterisks mark the amino acids likely contributing to the inability of CeAhR to bind big ligands. Amino acids highlighted by an arrow were mutated for the investigation of the LBD function (panels **F**,**H**). I 3D models of the CeAhR (left) and the mAhR (right) PASB domains obtained by homology modeling, shown in a cartoon representation. Secondary structures attributed by DSSPcont are labeled according to the PAS domain nomenclature. The colored internal area (blue for CeAhR and yellow for mAhR) defines the molecular surface of the binding cavity identified by CASTp. In the CeAhR model, the amino acids protruding into the binding cavity (asterisks in panel **D**) are labeled and shown as blue sticks. The mAhR amino acids corresponding to those displayed in the CeAhR model, are labeled and shown as yellow sticks. Amino acids highlighted by an arrow were mutated for studying the LBD function (panels **F**,**H**). (**F**) AhR activity in BaP- or MNF-treated Cos7 cells transfected with either AHR-1, an AHR-1 with L363A and H365Q mutations (LBD mutant), or mouse AhR (mAhR), as well as AHA-1 and an XRE-driven luciferase. Boxplots show data of 3 experiments. Statistical analysis: 2-way ANOVA and Tukey’s multiple comparisons test. * *p*-value < 0.05 vs. wt, ^#^
*p*-value < 0.05 vs. DMSO. (**G**) Effect of rotenone on AhR activity in Cos7 cells transfected with AHR-1 (either wt or *ju145*) as well as AHA-1 and an XRE-driven luciferase. Boxplots show data of 3 experiments. Statistical analysis: 2-way ANOVA and Tukey’s multiple comparisons test. * *p*-value < 0.05 vs. wt, ^#^
*p*-value < 0.05 vs. DMSO. (**H**) Effect of rotenone on AhR activity in Cos7 cells transfected with either AHR-1, AHR-1 with L363A and H365Q mutations (LBD mutant), or mouse AhR (mAhR). Boxplots show data of 3 experiments. Statistical analysis: 2-way ANOVA with Tukey’s multiple comparisons test. * *p*-value < 0.05 vs. wt/AHR-1, ^#^
*p*-value < 0.05 vs. DMSO/EtOH.

**Figure 7 antioxidants-11-00613-f007:**
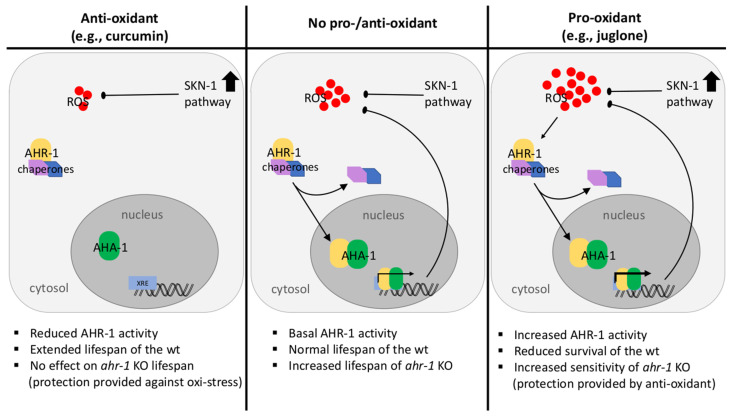
Proposed model of the AHR-1 signaling pathway in *C. elegans* response to pro- and antioxidants. In “normal” conditions (middle panel), AHR-1 is activated by intra-cellular ROS. This leads to the shedding of the chaperones from cytosolic AHR-1 and the subsequent nuclear translocation of AHR-1. In the nucleus, AHR-1 forms a heterodimer with the AHR nuclear translocator (AHA-1) and bind to XREs of target genes, which in turn leads to the reduction in intracellular ROS levels. In these conditions, loss of *ahr-1* function leads to an increased lifespan. In the presence of antioxidants (left panel) the intra-cellular ROS concentrations are low leading to AHR-1 residing in the cytoplasm, bound by its cofactors. The inhibition of the basal AHR-1 activity leads to an increased lifespan. In the presence of pro-oxidants (right panel) AHR-1 is activated by excessive ROS, which results its nuclear translocation, the AHR-1–AHA-1 heterodimer formation and the initiation of target gene transcription. In the *ahr-1* KO, the decreased detoxification of ROS through AHR-1-induced target genes leads to an accumulation of ROS and renders the *ahr-1* KO susceptible.

**Table 1 antioxidants-11-00613-t001:** List of genes from the microarray analysis.

Strongest Over-/Under-Expressed Genes by Curcumin in an *ahr-1*-Dependent Manner					
Gene/Sequence Name	Gene Class ^a^	Molecular Function ^a^	logFC ^b^	adj. *p*-Value ^c^	Selected Modulators ^a^
H43E16.1	unknown	unknown	1.53	0.021	bacterial infection, quercetin, rotenone, aging, *nuo-6(qm200)*
*numr-1*	Nuclear localized metal responsive	unknown	1.42	0.034	bacterial infection, quercetin, *spg-7* RNAi, *isp-1(qm150)*, *nuo-6(qm200)*, aging
*mul-1*	Mucin-like	unknown	1.34	0.029	resveratrol, bacterial infection, *spg-7* RNAi, rotenone, paraquat, indole, *isp-1(qm150)*, *nuo-6(qm200)*
*oac-14*	O-acyltransferase homolog	transferase activity, transferring acyl groups other than amino-acyl groups	1.24	0.085	bacterial infection, quercetin, tryptophan, rotenone, paraquat, indole, *nuo-6(qm200)*
F58B4.5	unknown	unknown	1.21	0.030	resveratrol, quercetin, *spg-7* RNAi, tryptophan, *isp-1(qm150)*, *nuo-6(qm200)*, paraquat indole,
*comt-4*	Catechol-O-methyl-transferase	O-methyltransferase activity	1.17	0.030	pathogenic bacteria, *ahr-1(ju145),* quercetin, rotenone, paraquat, *isp-1(qm150)*, *nuo-6(qm200)*, indole, aging
F09C8.1	Ortholog of human phospholipase B1	Phospholipase activity; hydrolase activity, acting on ester bonds	1.08	0.030	*ahr-1(ju145)*, bacterial infection, quercetin, rotenone, paraquat, *isp-1(qm150)*, *nuo-6(qm200)*, aging
*ugt-48*	UDP-glucuronosyl-transferase	calmodulin binding, glucuronosyltransferase activity, UDP-glycosyltransferase activity, transferase activity, transferring hexosyl and glycosyl groups	1.07	0.034	bacterial infection, rotenone, aging
*cyp-13A5*	Cytochrome P450	monooxygenase activity, metal ion binding, heme binding, oxidoreductase activity	1.05	0.057	bacterial infection, quercetin, *spg-7* RNAi, tryptophan, *isp-1(qm150)*, *nuo-6(qm200)*, indole
T19C9.8	unknown	unknown	0.96	0.084	bacterial infection, quercetin, *spg-7* RNAi, tryptophan, paraquat, *isp-1(qm150)*, *nuo-6(qm200)*
*lys-7*	Lysozyme	unknown	−1.54	0.084	bacterial infection, aging, rotenone, *isp-1(qm150)*, *nuo-6(qm200)*
*cyp-35A5*	Cytochrome P450	monooxygenase activity, metal ion binding, heme binding, oxidoreductase activity, steroid hydroxylase activity	−1.08	0.087	bacterial infection, *ahr-1(ju145)*, aging, tryptophan, rotenone, *isp-1(qm150)*, *nuo-6(qm200)*
C14A4.9	unknown	unknown	−0.55	0.079	bacterial infection, quercetin, rotenone, indole
**Genes Regulated in the Same Way by Curcumin or *ahr-1* Depletion**					
**Gene/Sequence Name**	**Gene Class ^a^**	**Molecular Function ^a^**		**LogFC ^d^**	**LogFC ^e^**
*slc-17.5*	Solute carrier homolog	transmembrane transporter activity		0.64	0.46
*ugt-9*	UDP-glucuronosyl-transferase	glucuronosyltransferase activity, transferase activity, transferring hexosyl groups		0.57	0.41
*nhr-239*	Nuclear hormone receptor	metal ion binding, zinc ion binding, transcription factor activity, sequence-specific DNA binding		0.43	0.39
*ugt-29*	UDP-glucuronosyl-transferase	glucuronosyltransferase activity, transferase activity, transferring hexosyl and glycosyl groups		0.36	0.34
C14A4.9	unknown	unknown		−0.55	−0.63

**^a^** Extracted from Wormbase; **^b^** LogFC: logarithmic fold change; **^c^** Adj. *p*-value: adjusted *p*-value; **^d^** logarithmic fold change wt curcumin vs. wt DMSO; **^e^** logarithmic fold change *ahr-1* DMSO vs. wt DMSO.

## Data Availability

Data are contained within the article and Appendix A.

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
