# Peer review of "Aryl Hydrocarbon Receptor-Dependent and -Independent Pathways Mediate Curcumin Anti-Aging Effects"

_antioxidants, 2022, doi:10.3390/antiox11040613_

Round 1
Reviewer 1 Report
In the reviewed work, the subtle mechanisms of the geroprotective action of such a well-known substance as curcumin were studied. The research topic is relevant. Research methods are modern, the interpretation of the results is correct
Author Response
We really appreciated the positive feedback of this reviewer.
Reviewer 2 Report
The aryl hydrocarbon receptor (AhR) and curcumin have evolutionarily conserved effects on aging. Vanessa Brinkmann et al. investigated whether and how the AhR mediates the anti-aging effects of curcumin across species.
In general the manuscript contain relevant paragraphs that have been discussed. The selection of bibliography is appropriate to the content of the manuscript.
Finally, regarding methodology, authors refer about statistics thus the readers can make assumptions regarding the quality and the confidence of the results and reasonability of consideration of the authors.
This study was conducted clearly and provided some interesting results about the function of AhR. The manuscript is very enjoyable to read, but after close evaluation of the paper I suggest revision according to the next point:
1. Indicate the element of novelty.
Author Response
We thank the reviewer for this important comment. We have now emphasized the element of novelty through the Discussion and Conclusions chapter of our manuscript and included a schematic figure (New Figure 7) summarizing our findings.
Reviewer 3 Report
The aryl hydrocarbon receptor (AhR) is a protein, encoded in humans by the AHR gene, which acts as a transcription factor that recognizes aromatic hydrocarbons.
Curcumin, the main biologically active compound extracted from Curcuma longa, is a powerful antioxidant. A broad spectrum of studies has demonstrated its ability to induce numerous biological and pharmacological effects. In this work, the authors investigated whether and how AhR mediates the anti-aging effects of curcumin across species, using a combination of in vivo, in vitro and in silico analyses.
The work is very complex, but well structured. Different types of analyses are reported and this explains the contribution of different authors.
The experimental part is conducted satisfactorily.
However, some changes are required.
Line 102: Authors should provide a brief explanation of Nrf2/SKN-1. For example: As is known, the transcription factor Nrf2, present in mammalian cells and its homolog SKN-1 of C. elegans are transcription factors that play a fundamental role in the response to oxidative stress, cell homeostasis and life span of the organism (with relative reference).
To facilitate the reading of the manuscript, given the large experimental part, the authors could create a figure that summarizes the pathway followed.
References should be reported as indicated in the Instructions for Authors.
Author Response
We thank the reviewer for the constructive comments. As suggested, we have now:
- included a short sentence briefly describing Nrf2/SKN-1 when it is first mentioned in the Introduction;
- included a schematic figure (New Figure 7) summarizing our findings;
- formatted the references as per journal guidelines.